# Development and accuracy assessment of molecular markers associated with crown rust resistance genes in oat

Rawnaq N. Chowdhury[1], Raja Sekhar Nandety[2], Belayneh Admassu Yimer[3], Jason Fiedler[2], Suraj Sapkota[3], Kathy Esvelt Klos[3]*

1 Oak Ridge Institute for Science and Education (ORISE) Research Participant, Small Grains and Potato Germplasm Research Unit, Aberdeen, Idaho, United States of America, 2 USDA-ARS, Edward T. Schafer Agricultural Research Centre, Fargo, North Dakota, United States of America, 3 USDA-ARS, Small Grains and Potato Germplasm Research Unit, Aberdeen, Idaho, United States of America

* kathy.klos@usda.gov

## Abstract

Crown rust caused by *Puccinia coronata* Cda. f. sp. *avenae* P. Syd. (*Pca*) is considered the most destructive disease of oat, causing yield and grain quality losses. Over a hundred crown rust race-specific resistance genes have been identified, but the history of cultivar development has left the identity of *Pc* resistance genes elusive. Closely linked molecular markers may be used to identify the carrier status of a particular *Pc* resistance allele in any given oat line. However, elevated false positive rates could lead to misidentifying carriers, potentially excluding valuable genetic material from breeding programs. There are very few studies that examine the reliability of gene molecular markers in a diverse genetic background. In this study, molecular markers with genotype data from the T3/Oat database and map data from GrainGenes, which indicated linkage to *Pc* genes, were evaluated for their predictive potential. A panel of non-carrier lines for *Pc* genes was identified using phenotype data downloaded from T3/Oat database and pedigree records from Pedigrees of Oat Lines database. The false positive rate of the markers was calculated as the percentage of non-carriers possessing the allele associated with the *Pc* gene. Using the available map information, thirty SNPs associated with 15 *Pc* genes were selected and assessed for their predictive capabilities. Eight out of the thirty markers, linked to seven *Pc* genes, showed potential in predicting carrier status with a false positive rate of ≤25% in non-carrier lines. Particularly, markers for *Pc38* and *Pc68* perfectly corresponded to carrier status across all lines. Furthermore, validation of published predictive markers for four *Pc* genes in this non-carrier panel demonstrated consistency with published data, with only a 6–17% genotyping error observed for three markers. Such markers have potential to identify *Pc* genes present in germplasm with resistance of unknown derivation, thereby enhancing the marker assisted selection process for oat breeding.

**Data availability statement:** All relevant data are within the paper and its Supporting Information files.

**Funding:** The research at USDA-ARS in Aberdeen, ID was supported with base fund Project No.: 2050-21000-034-000D. The funders had no role in study design, data collection and analysis, decision to publish, or preparation of the manuscript.

**Competing interests:** The authors have declared that no competing interests exist.

## Introduction

The human population is expected to reach nine billion by 2050 and securing food for this large population is a challenge [1]. To meet the demand of such a large population, there is an urgent need to increase food production. One possible way to increase food production is the integration of modern technologies, such as marker-assisted selection (MAS) in crop improvement programs. MAS is the process of utilizing molecular markers, such as simple sequence repeat (SSR) and single nucleotide polymorphism (SNP), that are linked with traits of interest to select desirable individuals. As compared to classical phenotypic selection, MAS is easier, faster, cheaper, and more efficient [2]. However, there are multiple factors that determine the effectiveness of MAS in plant breeding, including the reliability and accuracy of quantitative trait loci (QTL) mapping and the linkage disequilibrium between the marker and the QTL [1–3]. Most importantly, effectiveness of MAS depends on two pairs of correlated variables: trait vs QTL and QTL vs marker [1]. MAS has been commonly used in plant breeding to select for many economically important traits, including disease resistance [4], high yield [5], abiotic stress tolerance, and quality [6].

It is a proven fact that MAS can accelerate plant breeding work; however, little information is available on factors affecting the success of MAS in crop improvement. False positive and false negative rates are among many quality matrices that determine the effectiveness of markers used in MAS. The false positive rate is the proportion of known non- carriers of a trait incorrectly classified as positive carriers, whereas the false negative rate is the converse [1]. False positives and false negatives could lead to misidentifying carrier germplasm, potentially excluding valuable genetic material from breeding programs or inadvertently incorporating undesired traits.

Crown rust disease of oat, caused by the pathogen *Puccinia coronata* f. sp. *avenae*, causes both yield and grain quality losses [7]. Host resistance is the most preferred method to control oat crown rust and more than 100 resistance genes (*Pc* genes) have been discovered so far [8]. Of the > 100 *Pc* genes discovered, only 11 genes (*Pc38, Pc39, Pc45, Pc53, Pc54, Pc58a, Pc68, Pc71, Pc91, Pc96,* and *Pc98*) have been mapped [8–18] and subsequently positioned on the oat consensus map [19]. Some additional *Pc* genes have been assigned to a linkage group, but information about physical location in the oat genome is unavailable. The lack of information about the chromosomal location of most *Pc* genes has limited the utilization of genomic tools in oat breeding. Molecular markers closely linked to *Pc* genes provide a quick way to identify germplasm with potential known or novel crown rust resistance genes. Functional markers also known as precision markers or diagnostic markers enable efficient, quick characterization and screening of germplasm for allelic diversity with accuracy since they are not subjected to recombination [20]. Close genomic association of diagnostic markers with the gene of interest serves to increase the selection efficiencies compared to other markers for variety development. Unfortunately, only four of the over 100 *Pc* genes have published diagnostic molecular markers. Therefore, more studies are needed to determine the location of *Pc* genes and develop user-friendly molecular markers to be used in breeding programs.

Recent advancements in oat genomics have greatly facilitated the development and use of molecular markers in crop improvement. For example, the *A. sativa* genome assembly v1 was recently published by PepsiCo in partnership with several public and private research organizations for use in open-source applications (https://wheat.pw.usda.gov/GG3/grain-genes.downloads/oat-ot3098-pepsico). These developments have created an opportunity for determining the genomic locations of *Pc* genes and subsequently develop *PC*R-based diagnostic markers for MAS. The objective of this study was to evaluate PCR allele competitive extension (PACE) assays developed for SNPs linked with *Pc* genes based on their false positive rates in a non-carrier panel of oat lines.

## Materials and methods

### Oat germplasm

The present study used 92 oat lines selected from the Collaborative Oat Research Enterprise (CORE) association mapping panel, as described by Klos et al. [33], along with 31 *Pc* differential lines obtained from the Cereal Disease Laboratory in St. Paul, MN. The 92 oat lines were selected based on their susceptibility to crown rust when evaluated in a controlled environment using ten *Pca* races [21], as well as, in field evaluations in six locations for the 2010–2011 years [21]. Lines were classified as susceptible if they met the following criteria: an average infection type (IT) score of ≥ 3 across all growth chamber tests (covering ten *Pca* races), an infection reaction (IR) score of ≥ 0.7, and a severity percentage of ≥ 40 across all locations and years (excluding Winnipeg 2011) as reported in the Oat Toolbox database (https://oat.triticeaetoolbox.org/). In addition, lines were considered non-carriers if their pedigree showed no indication of *Pc* genes, according to the Pedigrees of Oat Lines database (https://triticeaetoolbox.org/POOL/). Allele frequencies of molecular markers were determined using genotype data from the public database on T3 (https://oat.triticeaetoolbox.org/). The selected non-carrier lines were genotyped using both the Illumina 6K (S1 Table in S File) and 3K (USDA_SoywheOatBar-3K) chips (S2 Table in S File) at the USDA–ARS Genotyping Laboratory in Fargo, ND, following the methodology described by Tinker et al. [22]. To validate the PACE assays developed in this study, a separate set of 31 non-carrier oat lines was used, 20 of which were also included in the original group of 92 oat lines.

### Selection of crown rust resistance genes and linked SNPs

Fifteen *Pc* genes were selected based on the availability of linkage map data. These include three genes (*Pc53, Pc54,* and *Pc96*) mapped using a SNP array, seven genes (*Pc38, Pc46, Pc48, Pc50, Pc58a, Pc68,* and *Pc71*) mapped using RFLP markers, four genes (*Pc35, Pc62, Pc63,* and *Pc55*) reported as either linked or allelic to other mapped *Pc* genes, and one, *Pc94*, mapped using SCAR markers.

For *Pc* genes mapped using a SNP array, the markers could be directly placed to the consensus map [19]. For example, *Pc53* was mapped to the linkage group Mrg08 (Chr. 2D) of the oat consensus map at 82.4 cM. Consequently, SNPs in this region with available genotype data from the oat CORE collection were downloaded and used for the analysis. Positions on the consensus map were inferred from the locations of linked RFLP markers for *Pc* genes mapped using RFLP markers, and potential SNP markers within those regions were identified. For example, Wight et al. [16] mapped *Pc38* to a region homologous to the KO (Kanota/Ogle) group 17 in two F$_{2:3}$ populations using RFLP markers. The KO group 17 aligns with Mrg02 (Chr. 7D) of the oat consensus map. Consequently, these RFLP markers were assigned to Mrg02 (Chr. 7D). Four *Pc* genes, reported as either linked or allelic to other mapped *Pc* genes, were arbitrarily assigned to the same map location as their linked counterparts. For example, *Pc35* was reported to be linked with *Pc54*, which was mapped to Mrg02 (Chr. 7D) of the oat consensus map. Thus, the same genomic region was used for *Pc35*. *Pc94*, which was mapped using SCAR markers, could not be positioned on the oat consensus map due to the absence of shared markers. The four SNP sequences for *Pc94* reported by Chong et al. [13] were used in this study to develop the PACE assay. The criteria for SNP identification considered the genomic regions on both sides of the most probable location of

each *Pc* gene on the oat consensus map, as well as the availability of SNP genotype data for *Pc* differential lines and non-carrier lines within those regions.

In addition, four *Pc* genes (*Pc39, Pc45, Pc91, Pc98*) for which diagnostic markers were previously developed by various authors [14,17,18,23] were included in the present study for the purpose of comparing the present assay with previously published assays.

### Determining false positive rates of selected SNP markers

The reliability of the markers with respect to the diagnosis of *Pc* gene carrier status was determined by calculating the false positive rate estimated for each SNP marker using genotype data of the 31 *Pc* differential and the 92 non-carrier lines. The allele carried by each *Pc* gene differential for each SNP corresponding to the various *Pc* genes was determined. The frequency of that allele in the non-carrier panel of susceptible lines was estimated to determine the false positive rate. For SNP combinations, the two SNPs were called together to determine the false positive rate (%). The alleles carried by each *Pc* gene differential for the two SNPs, corresponding to their respective *Pc* genes, were analyzed together to identify the presence of the respective *Pc* gene.

The false positive rate (%) was calculated by the following formula:

$$False\ positive\ rate\ (\%) = (yi/Yi)100$$

where $y_i$ = number of the non-carrier lines carrying the *Pc* gene corresponding to the allele; $Y_i$ = total number of non-carrier lines.

Between one and four SNP markers with the lowest false positive rates, corresponding to each of the 15 *Pc* genes, were selected for further PACE marker design (S4 Table). No SNPs were selected for *Pc48* due to the high false positive rates observed in the available SNPs (ranging from 41% to 82%, S4 Table).

### PACE marker design and assay development

PACE markers were developed at the USDA-ARS Small Grain Genotyping Center, Fargo, ND and 3CR Bioscience (Essex, UK) (www.3crbio.com/free-assay-design). PACE markers were designed from the sequences of selected SNPs using the free assay design service by 3CR Bioscience [Essex, UK] (www.3crbio.com/free-assay-design) and/or using polymarker (http://www.polymarker.info), an automated bioinformatics pipeline for SNP assay development. The PepsiCo oat genome OT3098v2 was used as a reference in designing the markers. Flanking sequences of the SNP markers (~100 bp upstream and downstream) were used for primer design. The automated design feature of polymarker was used to design the forward and reverse primers for each SNP, and primer sequences were ordered through IDT (https://www.idtdna.com). The sequence information of all thirty PACE primers is documented in the S6 Table in S File.

PACE assay evaluation and validation was done at the USDA-ARS Small Grains and Potato Germplasm Research Unit, in Aberdeen, ID and at the USDA-ARS Small Grain Genotyping Center, in Fargo, ND. The two forward primers and the common reverse primer were maintained at stock concentrations of 100 μM and PACE genotyping was assayed with the co-dominant PACE genotyping system (3cr Bioscience, https://3crbio.com/product/pace). A 100-μl assay mix was prepared from a mixture of 12 μl each of the forward primers, 30 μl of the reverse primer, and 46 μl of DE*PC* water. The final 10-μl reaction mixture was comprised of 5 μl of 2×PACE Genotyping Master Mix (3CR Bioscience, Essex, U.K.), 2.86 μl of DEPC water, 0.14 μl assay mix, and 2 μl of template DNA. Cycling conditions were 94° C for 15 min, 10 cycles of 94°C for 20 s and 65° C for 60 s with an annealing temperature decrement of 0.8°C per cycle, and 30 cycles of 94°C for 20 s and 57°C for 60 s (Bio-Rad, Hercules, CA). The PACE assays were optimized for each group of markers by adjusting the number of cycles, annealing temperature, and/or concentrations of the forward primers. The allele call information was deduced based on fluorescence data as an end-point genotyping using Roche Lightcycler 480 software and/or assays

were read for FAM and HEX fluorescence with a CFX96 real-time PCR detection system. Assay evaluation was done using DNA of the respective *Pc* differential line corresponding to the *Pc* genes (positive control) and a universally susceptible oat cultivar 'Ajay' as a non-carrier (negative control).

## Assay validation

Assay validation was conducted using DNA from the assay validation panel, which included 31 non-carrier lines and 21 carrier lines (S3 Table in S File). The master mix, PCR cycle conditions, and allele detection were the same as described above.

## Results

### False positive rates of SNP markers in non-carrier oat lines

The study identified four to 35 SNPs closely associated with each of the 15 *Pc* genes (Table 1). Three *Pc* genes (*Pc53, Pc54,* and *Pc96*), which were mapped using the SNP array, had 13–18 closely linked SNPs. The RFLP markers used to map *Pc38, Pc46, Pc48, Pc50, Pc58a, Pc68,* and *Pc71* were placed on the oat consensus map, aiding in the identification and selection of five to 35 linked SNP markers for each *Pc* gene (Table 1). However, the placement of *Pc46, Pc50,* and *Pc68* was uncertain, as RFLP markers closely associated with these genes were assigned to two different oat chromosomes, 7A or 3D. According to the original RFLP marker information, the oat stem rust resistance gene *Pg9*, linked with *Pc68*, which were both located on LG 4 of the Kanota/Ogle (KO) linkage map and the updated KO_4_12 map through comparative mapping. Portyanko et al. [24] identified a QTL '*Prq5*' on chromosome 7A, associated with *Pc46, Pc50,* and *Pc68*, which Acevedo et al. [25] reported to correspond to homoeologous linkage groups KO6 and KO4. Five and three of

**Table 1. Consensus map position of selected *Pc* genes with the linkage map location window within SNPs were identified, the number of SNP markers evaluated, and the range of false positive rates (%).**

| *Pc* Gene | Original markers | Linkage group[a] | Chromosome[b] | Position (cM) | Number of SNP[c] | False positive rate (%) |
|---|---|---|---|---|---|---|
| *Pc53* | SNP | Mrg08 | 2D | 62.4 - 109.4 | 13 | 8-79 |
| *Pc54* | SNP | Mrg02 | 7D | 30 - 70.8 | 17 | 23-63 |
| *Pc96* | SNP | Mrg02 | 7D | 72.7 - 87.3 | 18 | 11-98 |
| *Pc38* | RFLP | Mrg02 | 7D | 30 - 87.9 | 35 | 0-81 |
| *Pc46* | RFLP | Mrg12 | 7A | 7.5 - 39.4 | 14 | 14-80 |
| | | Mrg19 | 3D | 71 - 80.5 | 5 | 0-96 |
| *Pc48* | RFLP | Mrg20 | 4A | 84.2 - 156.7 | 10 | 41-82 |
| *Pc50* | RFLP | Mrg12 | 7A | 7.5 - 39.4 | 14 | 28-80 |
| | | Mrg19 | 3D | 71- 80.5 | 5 | 26-58 |
| *Pc58a* | RFLP | Mrg02 | 7D | 0 - 17.2 | 6 | 32-34 |
| *Pc68* | RFLP | Mrg12 | 7A | 7.5 - 39.4 | 14 | 28-80 |
| | | Mrg19 | 3D | 71 - 80.4 | 5 | 0-93 |
| *Pc71* | RFLP | Mrg05 | 6A | 130.9 - 138.8 | 5 | 13-64 |
| *Pc55* | Segregation | Mrg11 | 4C | 3.7 - 40.9 | 9 | 19-98 |
| *Pc62* | Segregation | Mrg02 | 7D | 71 - 87.9 | 18 | 31-98 |
| *Pc63* | Segregation | Mrg02 | 7D | 71 - 87.9 | 18 | 13-86 |
| *Pc35* | Segregation | Mrg02 | 7D | 30 - 70.8 | 17 | 23-63 |
| *Pc94* | SCAR | NA | NA | NA | 4 | NA |

[a] Linkage groups on the consensus linkage map of [19].

[b] Based on new chromosome numbering system for oats (*Avena* spp.) developed by [40].

[c] Number of SNPs evaluated for MAF in the 92 non-carrier oat panel.

these RFLP markers from the KO_4_12 linkage groups were assigned to linkage groups Mrg19 (Chr. 3D) and Mrg12 (Chr. 7A) of the oat consensus map, respectively. Consequently, SNPs on both chromosomes were searched and selected, resulting in the identification of 14 SNPs from chromosome 7A and five SNPs from chromosome 3D for these three genes (Table 1). The location of *Pc71* was uncertain as well, possibly located on Chr. 4D or 6A. Chr. 6A was searched to identify SNPs in this study (Table 1).

Based on the false positive rates among SNP markers linked to the 15 *Pc* genes (S4 Table in S File), the genes were categorized into four groups. Three *Pc* genes had SNP markers with false positive rates of less than 10% among the 92 non-carrier lines (Table 2). SNP markers associated with *Pc38* and *Pc68* achieved an ideal score of 0% false positive rate, indicating a high potential for correctly classifying uncharacterized germplasm with oat crown rust resistance. A SNP marker located 2.4 cM distal to *Pc53* exhibited a false positive rate of 8% (Table 2). Seven *Pc* genes had ten closely linked SNP markers with false positive rates ranging from 10% to 25% (Table 2). In contrast, two *Pc* genes showed false positive rates of 25% or higher for the closely linked SNPs, despite having more precise map locations (Table 3). SNP markers in this group are more likely to lead to the inadvertent discarding of valuable breeding germplasm, highlighting their classification errors in non-carrier lines.

**Table 2. Performance statistics of SNP markers with false positive rate(s) ≤25% observed per Pc gene.**

| *Pc* Gene | Name of linked SNP | Position(cM)[a] | False positive rate (%) in the non-carrier panel[b] | |
|---|---|---|---|---|
| | | | ≤10% | 10-25% |
| *Pc38* | GMI_GBS_92811 | 30.1 | 0 | – |
| *Pc68* | avgbs_218381[c] | 32.1 | 0 | – |
| | GMI_ES01_lrc9332_388 [c] | 29.5 | 9 | – |
| *Pc53* | GMI_ES01_c28412_66 | 82.4 | 8 | – |
| *Pc35/Pc54*[d] | GMI_ES15_c14565_407 | 51.3 | – | 23 |
| *Pc46* | GMI_ES_LB_9851 | 7.5 | – | 14 |
| *Pc55* | GMI_ES01_c719_992 | 3.7 | – | 20 |
| | GMI_ES01_c1793_450 | 12.3 | – | 22 |
| | GMI_ES_CC4633_339 | 9.8 | – | 19 |
| *Pc63* | GMI_E01_c2095_229 | 85.2 | – | 13 |
| | GMI_ES01_c8043_192 | 85.2 | – | 16 |
| *Pc71* | GMI_DS_LB_5942 | 130.9 | – | 23 |
| | GMI_GBS_21750 | 135.5 | – | 13 |
| *Pc96* | GMI_ES02_c23895_331 | 87.3 | – | 18 |

[a]cM refers to the centimorgan position on the consensus linkage map of [19].

[b]False positive rate (%) based on the frequency of carrier allele in a panel of 92 non-carrier oat lines.

[c]marker chosen from 3K SNP chip genotype data.

[d]*Pc35* and *Pc54* had the same genotype in all markers on Mrg02.

**Table 3. Performance statistics of SNP markers with false positive rate(s) ≥25% observed per *Pc* gene.**

| *Pc* Gene | Name of linked SNP(s) or SNP pairs | Position (cM)[a] | False positive rate (%) in the non-carrier panel[b] |
|---|---|---|---|
| *Pc58a* | GMI_DS_LB_6614 | 10.8 | 32 |
| *Pc48* | GMI_ES01_c1107_182 | 84.2 | 41 |

[a]cM refers to the centimorgan position on the consensus linkage map of [19].

[b]False positive rate (%) based on the frequency of carrier allele in a panel of 92 non-carrier oat lines.

Four *Pc* genes had eight single SNP markers that resulted in high false positive rates (Table 4). However, when SNP pairs were used in combination, the genotypes showed reduced false positive rates. For example, the SNPs GMI_ES17_ c3973_587 and GMI_ES22_c4822_80, both linked to *Pc50*, had individual false positive rates of 38% and 61%, respectively. When used together, the false positive rate dropped to 9%. Similar reductions were observed for *Pc55, Pc62*, and *Pc71,* with false positive rates decreased to 14%, 15 and 7%, respectively, when SNP pairs were analyzed together (Table 4). This indicates that combined SNP assays can potentially increase the probability of selecting the desired gene.

### PACE assay development and validation

The design of PACE markers was successful for 27 out of the 30 selected SNP markers (see S6 Table). Of these, 21 PACE markers effectively differentiated the alleles in the differential resistant lines (*Pc35*, *Pc38*, *Pc46*, *Pc50*, *Pc53*, *Pc54*, *Pc55*, *Pc58a*, *Pc62*, *Pc68*, *Pc94* and *Pc96*) from the non-carrier alleles in the susceptible lines (S6 Table). However, six markers were either monomorphic or had a high rate of no calls (S6 Table). Additionally, 14 of the 21 successful markers were discarded due to a high false positive rate (≥ 25%) during validation with an oat panel [18]. For instance, the marker 'GMI_GBS_32754_PACE' developed for *Pc35* exhibited a 29% false positive rate in the validation panel and was therefore excluded from this study (S8 Table in S File). Although a marker with a false positive rate above zero will incorrectly classify some non-carrier lines as carriers, this study considered a false positive rate of 25% or higher to be excessively high. Such a rate poses a substantial risk of incorrect conclusions, thereby limiting the practical utility of the developed markers. Therefore, PACE markers with ≥25% false positive rate were not reported in this study.

The remaining eight markers, which were predictive of seven *Pc* genes: *Pc38, Pc46, Pc53, Pc55, Pc58a, Pc71,* and *Pc96,* showed a false positive rate of less than 25% (Table 5 and S7 Table in S File). Notably, the markers GMI_ GBS_92811_PACE and avgbs_218381_PACE demonstrated exceptional accuracy, with no false positives in predicting *Pc38* and *Pc68*, respectively, during validation. The other PACE markers predicted their respective genes with a false positive rate ranging from 13% to 22% (Table 5 andS7 Table in S File). Overall, this study successfully developed PCR-based molecular markers for nearly half (seven out of 15) of the *Pc* genes with mapped locations, maintaining a false positive rate below 25% (Table 5).

### Comparison of current assay with previous assays

The present study validated all ten PACE markers except for one, Avgbs2_172881.1, a newly developed GBS SNP marker representing *Pc45*, where the alternative allele was not identified in the *Pc45* differential line. PACE markers corresponding to *Pc39, Pc45,* and *Pc91* showed 100% similarity in the genotypic data within the shared subset panel. However, three PACE markers for *Pc*98 exhibited a 6–17% error rate in the genotyping data (Table 6 and S5 Table in S File).

**Table 4. Performance statistics of SNP markers and SNP pairs with reduced false positive rate(s) observed per *Pc* gene.**

| *Pc* Gene | Name of linked SNP(s) or SNP pairs | Position (cM)[a] | False positive rate (%)[b] | |
|---|---|---|---|---|
| | | | individual markers | paired markers |
| *Pc50* | GMI_ES22_c4822_80/GMI_ES17_c3973_587 | 17/25 | 61/38 | 9 |
| | GMI_ES01_c10026_184/GMI_ES17_c3973_587 | 20.3/25 | 71/38 | 9 |
| *Pc55* | GMI_ES01_c719_992/GMI_ES_CC4633_339 | 3.7/9.8 | 20/19 | 14 |
| *Pc62* | GMI_ES01_c20384_216/GMI_ES02_c23895_331 | 87.3 | 31/18 | 15 |
| *Pc71* | GMI_DS_LB_5942/GMI_GBS_21750 | 130.9/135.5 | 23/13 | 8 |

[a]cM refers to the centimorgan position on the consensus linkage map of [19].

[b]False positive rate (%) based on the frequency of carrier allele in a panel of 92 non-carrier oat lines.

**Table 5. False positive rates of PACE assays in 31 non-carrier oat lines.**

| Pc Gene | PACE marker | False positive rate (%) [a] |
|---------|-------------|-----------------------------|
| Pc38 | GMI_GBS_92811_PACE | 0 |
| | GMI_ES02_c1532_592_PACE | 21 |
| Pc46 | GMI_ES_LB_9851_PACE | 13 |
| Pc53 | GMI_ES01_c28412_66_PACE | 12 |
| Pc55 | GMI_ES01_c719_992_PACE | 16 |
| Pc58a | GMI_DS_LB_6614_PACE | 19 |
| Pc68 | avgbs_218381_PACE | 0 |
| Pc96 | GMI_ES02_c23895_331_PACE | 22 |

[a]False positive rate (%) based on the frequency of carrier allele in the marker assay validation panel of 31 susceptible (non-carrier) oat lines.

**Table 6. False positive rates observed in this study for published KASP markers predictive of four Pc genes in presumably non-carrier lines.**

| Pc gene | Published marker | False positive rate (%) | | Shared non-carrier lines | Genotype similarity (%) |
|---------|------------------|-------------------------|---|--------------------------|-------------------------|
| | | Published data | Present study[a] | | |
| Pc39 (Zhao et al. 2020) | GMI_ES15_c6153_392 | 3 | 3 | 18/74 | 100 |
| Pc45 (Kebede et al. 2018) | I05-0090-KOM13c1 | 0 | 5 | 26/71 | 100 |
| | I05-0874-KOM16c1 | 0 | 0 | 26/71 | 100 |
| | Avgbs2_172881.1 | 1 | – | – | – |
| Pc91 (Gnanesh et al. 2013) | oPt-0350-KOM4C2 | 0 | 0 | 5/15 | 100 |
| | oPt-0350-KOM5C1 | 0 | 3 | 5/15 | 100 |
| | oPt-0350-KOM5C2 | 0 | 0 | 5/15 | 100 |
| Pc98 (Zhao et al. 2020) | GM_DS_LB_7494_kom370 | 5 | 3 | 18/79 | 94 |
| | Avgbs2_153634.1.59_kom410 | 24 | 3 | 18/79 | 89 |
| | GMI_ES15_c794_171_kom364 | 24 | 36 | 18/79 | 83 |

[a] Out of 31 susceptible (non-carrier) oat lines.

## Discussion

The present study outlines criteria for assessing the reliability of both published and unpublished SNP markers associated with 19 Pc genes in oats, enabling the design of an optimal marker system. A total of 19 markers, for 13 Pc genes, demonstrated the ability to predict carrier status with a false positive rate of ≤25% in non-carrier lines. PACE markers for seven Pc genes were developed and validated in this study. Additionally, the study discussed the importance of evaluating the accuracy and reliability of specific markers, the impact of marker type on selection accuracy, the usefulness of the false positive rate in identifying candidate markers for transfer to a different marker system, and the current limitations for marker development in oats.

Identifying markers that reliably indicate the presence or absence of genes in any given line, regardless of allelic diversity, is crucial for oat breeding. Currently, there is limited literature on designing reliable markers and criteria for judging marker quality is scarce. Typically, closely linked markers identified in mapped Pc genes are used to determine the presence of these genes in diverse germplasm panels [14,17,18,23]. However, challenges arise when such markers are applied to new populations, especially when one parent's allele status is unknown. Markers that are informative in a specific mapping population may perform poorly when deployed in new breeding programs. Therefore, methods to characterize markers are needed to identify those at risk of giving misleading results, along with stringent estimates of false positive

rates for these markers. For example, Zhao et al. [18] designed 20 KASP markers to map *Pc98* in two populations, identifying three of these markers as diagnostic. These markers were compared in this study, revealing slight variations in false positive rates among non-carrier panels despite using different subsets of the same CORE panel. Factors such as different assay chemistries or low call clarity could affect technical metrics, even when targeting the same polymorphism. Quantifying these markers with false positive rates would clearly demonstrate which marker is superior and how reliably these markers can be used in other breeding programs, thus greatly enhancing breeding efficiency.

Validating the existing closely linked SNP markers using accuracy matrices, such as the false positive rate [1], illustrates that there is no inherent advantage of one marker type over another in terms of selection accuracy. Although SNP markers from the RFLP and SNP linkage maps varied in their false positive rates, the SNP markers for *Pc38* and *Pc68* in this category had a perfect false positive score of 0%. Conversely, none of the SNPs from genes placed on the consensus map using segregation data (*Pc53, Pc55, Pc62,* and *Pc63*) achieved perfect false positive scores, which might mean that earlier linkage results were misleading. For instance, the map positions of *Pc62* and *Pc63* were based on their linked or allelic relationship with *Pc38* [26]. Additional linkage mapping would clarify the map locations of these genes. While it is possible to use SNP markers from linkage data, it is also likely that this study did not have access to sufficient SNPs in these linked regions to identify markers in high linkage disequilibrium (LD) with the resistance gene. Therefore, the choice of marker type has less impact on selection accuracy, and the best marker will be determined by its level of association with the target *Pc* gene.

In this study, the applied accuracy metrics describe the level of association between the marker and the target gene, as outlined by Platten et al. [1]. This ensures that the selected genes are genuinely present in the breeding material, leading to more reliable and precise breeding outcomes. Breeders can confidently select individual lines with specific *Pc* genes. However, markers with a high false positive rate pose a significant risk of misclassifying lines as carriers of *Pc* genes when they do not have the trait, potentially advancing materials that should be discarded. For example, when developing simple PCR-based markers for the stem rust resistance gene *Sr32* [27], marker-trait associations were validated using near-isogenic lines, an F3 segregating family, and four other *Sr32* translocations. However, the marker for *Sr32* was designed to target the recipient allele of the gene rather than the donor allele. As a result, there was a potential risk of misclassifying some non-carrier lines as carrier lines. In this study, PACE markers with false positive rates of ≤25% linked to *Pc* genes were selected, as reported by Zhao et al. [18], who identified two KASP markers (Avgbs2_153634.1.59_kom410 and GMI_ES15_c794_171_kom364) as predictive of *Pc98*, with a prediction accuracy of 76%. On the other hand, SSR markers *X3B042G11* and *X3B028F08*, associated with the stem rust resistance gene *Sr2*, were not recommended by Bernardo et al. [28] due to their high false positive rates (38% and 68%) in various wheat accessions lacking the target resistance genes. Markers with a non-zero false positive rate may find applications in fine-mapping or specific populations, but those with perfect scores are the most reliable for breeding purposes. Oat breeders can choose markers and decide on the acceptable error rate based on the specific objectives of their breeding program.

Uncertain gene locations complicate the process of identifying reliable markers and increase the risk of false positive and false negative associations. Without precise locations, selecting markers closely linked to the genes becomes problematic, often resulting in markers that are not in linkage disequilibrium with the genes, and thereby reducing their reliability. In this study, the locations of *Pc46*, *Pc50,* and *Pc68*, part of a large gene cluster with the stem rust resistance genes *Pg3* and *Pg9* [29] were uncertain. As mentioned above, *Pc*46 and *Pc*68 RFLP markers were placed to both Mrg19 (Chr. 3D) and Mrg12 (Chr. 7A) in the consensus map. Single nucleotide polymorphisms evaluated in this study with the best predictive potential were all on chromosome 7A (Mrg12), mapped to 7.5 cM and 32.1 cM on the consensus map. This finding aligns with previous studies that reported linkage between *Pg9* and *Pc*68 [30] and between *Pc*68 and *Pc*46 [31]. However, recent linkage mapping suggested that *Pc*46 may be on chromosome 3D (Mrg19) [Aaron Beattie, *personal communication*], consistent with the placement of potentially linked RFLP markers on chromosome 3D (Mrg19) [12,21,32] in the consensus map. Clearly, more work is needed to resolve the positions of *Pc46*, *Pc50*, and *Pc68*. It is also possible

that the *Pc46* differential line carries more than one crown rust resistance gene, as observed in *Pc58* [15] and *Pc50* [33]. Similarly, *Pc71*'s position was uncertain, with RFLP markers mapped to chromosomes 4D [34] and 6A [21] in the consensus map, rendering the exact location of this gene unclear. Although two SNP markers on chromosome 6A showed promise with a false positive rate of 13 and 23%, the development of PACE assays for these markers was unsuccessful. Further research is needed to identify reliable markers for *Pc71*, particularly on chromosome 4D.

The possibility that lines in the negative control panel were misclassified as not carrying *Pc* genes cannot be ignored. The susceptible negative control panel lines were selected based on seedling resistance data that were generated using a set of 10 isolates collected from both spring and winter oat growing regions. It is likely that this set of isolates was not informative of the absence of all *Pc* genes. In addition, the presence of *Pc* gene inhibitors may mask the resistance effects of some *Pc* genes resulting in misclassification of susceptible lines. For example, the suppression effect of *Pc38* on *Pc62* has previously been confirmed, as has the suppression of *Pc94* by *Pc38* [35–37]. However, the pedigree records of the negative control lines were obtained from the "Pedigrees of Oat Lines" POOL database [38,39], reducing the likelihood that any were carriers.

The performance of markers within a breeding program was incompletely characterized in this study due to the lack of information on expected false negative rates. Because of the limited information on the *Pc* gene carrier status of oat cultivars, too few known carrier lines were identified to obtain accurate estimates of false negative rates. For instance, among the seven *Pc* genes for which promising PACE markers were developed in this study, only three (*Pc38, Pc58a, Pc68*) had elite oat lines known to carry them (six, six, and five oat lines, respectively; S7 Table). These numbers were insufficient to conclusively estimate false negative rates. Previous researchers faced similar challenges, with only one to ten oat lines available for estimating false negative rates while developing diagnostic markers for *Pc39, Pc45, Pc91*, and *Pc98* [14,17,18,23]. An inadequate dataset does not lead to the rejection of markers but implies limited power to distinguish between them. Therefore, it is essential to report the total number of defined carrier and non-carrier lines in published results to demonstrate the accuracy of the outcomes.

In summary, the PACE markers developed in this study will allow the prediction of *Pc* gene carrier status in diverse germplasm with ≤25% false positives and are well-suited for high-throughput genotyping. This approach is expected to aid in the development of a robust marker system and underscores the need for further mapping of *Pc* genes. The PACE markers established in this study will be valuable tools for germplasm characterization and marker-assisted selection for crown rust resistance in oat.

## Supporting information

**S1 Table. List of susceptible oat lines (with pedigrees) used in SNP selection (genotyped with 6K chip) to estimate the false positive rate (%) of the *Pc* gene SNPs.**
(XLSX)

**S2 Table. List of susceptible oat lines (with pedigrees) used in SNP selection (genotyped with 3K chip) to estimate the false positive rate (%) of the *Pc* gene SNPs.**
(XLSX)

**S3 Table. Negative and positive control oat lines (with pedigrees) used as marker assay validation panel to estimate the false positive rate (%) of the *Pc* gene PACE markers.**
(XLSX)

**S4 Table. Performance statistics of SNPs (or SNP pairs) with the lowest false positive rate(s) observed per *Pc* gene.**
(XLSX)

**S5 Table. A list of SNP markers, designated ID, SNP ID and probes for KASP assays published for four *Pc* genes.**
(XLSX)

**S6 Table. A list of SNP markers, PACE markers, SNP information, primer information for PACE assays developed and validated for 15 *Pc* genes (except *Pc48*).**
(XLSX)

**S7 Table. Allele pattern of eight PCR allele competitive extension (PACE) markers for corresponding seven *Pc* genes validated in marker assay validation panel.**
(XLSX)

**S8 Table. Allele pattern of fourteen PCR allele competitive extension (PACE) markers for corresponding nine *Pc* genes validated in marker assay validation panel.**
(XLSX)

## Acknowledgments

All opinions expressed in this paper are the author's and do not necessarily reflect the policies and views of USDA, ARS, DOE, or ORAU/ORISE. The authors acknowledge the support of Mary Osenga at the Cereal Crops Research Unit in Fargo, ND, for genotyping the validation population with PACE assay.

## Author contributions

**Conceptualization:** Rawnaq N. Chowdhury, Belayneh Admassu Yimer, Kathy Esvelt Klos.

**Data curation:** Rawnaq N. Chowdhury, Raja Sekhar Nandety.

**Formal analysis:** Rawnaq N. Chowdhury.

**Funding acquisition:** Kathy Esvelt Klos.

**Investigation:** Rawnaq N. Chowdhury, Kathy Esvelt Klos.

**Methodology:** Rawnaq N. Chowdhury.

**Project administration:** Rawnaq N. Chowdhury, Kathy Esvelt Klos.

**Resources:** Raja Sekhar Nandety, Jason Fiedler, Kathy Esvelt Klos.

**Supervision:** Kathy Esvelt Klos.

**Writing – original draft:** Rawnaq N. Chowdhury.

**Writing – review & editing:** Rawnaq N. Chowdhury, Raja Sekhar Nandety, Belayneh Admassu Yimer, Jason Fiedler, Suraj Sapkota, Kathy Esvelt Klos.

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
