## [Decision Letter · Decision Letter 0]

PLOS ONE

Dear Dr. Chowdhury,

Thank you for submitting your manuscript to PLOS ONE. After careful consideration, we feel that it has merit but does not fully meet PLOS ONE’s publication criteria as it currently stands. Therefore, we invite you to submit a revised version of the manuscript that addresses the points raised during the review process.

**A**

Dear Dr. Chowdhury,

I received comments from the advisers on your manuscript "Development and accuracy assessment of molecular markers associated with crown rust resistance genes in oat" which you submitted to PlosONE. Based on reviewer comments and my careful assessment I have decided that your manuscript could be reconsidered for publication after minor improvements. When preparing a revised manuscript, you are asked to carefully consider the reviewer comments and submit a list of detailed and itemized responses to the comments.

With kind regards,

Dragan

We look forward to receiving your revised manuscript.

Kind regards,

Dragan Perovic, Ph.D

Academic Editor

PLOS ONE

“The research at USDA-ARS in Aberdeen, ID was supported with base fund Project No.: 2050-21000-034-000D”

Additional Editor Comments:

Dear Dr. Chowdhury,

I received comments from the advisers on your manuscript "Development and accuracy assessment of molecular markers associated with crown rust resistance genes in oat" which you submitted to PlosONE. Based on reviewer comments and my careful assessment I have decided that your manuscript could be reconsidered for publication after minor improvements. When preparing a revised manuscript, you are asked to carefully consider the reviewer comments and submit a list of detailed and itemized responses to the comments.

With kind regards,

Dragan

Reviewers' comments:

Reviewer's Responses to Questions

**Comments to the Author**

1. Is the manuscript technically sound, and do the data support the conclusions?

Reviewer #1: Yes

Reviewer #2: Yes

2. Has the statistical analysis been performed appropriately and rigorously?

Reviewer #1: Yes

Reviewer #2: Yes

3. Have the authors made all data underlying the findings in their manuscript fully available?

Reviewer #1: Yes

Reviewer #2: Yes

4. Is the manuscript presented in an intelligible fashion and written in standard English?

Reviewer #1: Yes

Reviewer #2: Yes

Reviewer #1: The manuscript ‘Development and accuracy assessment of molecular markers associated with crown rust resistance genes in oat’ by Chowdhury et al. is of topical interest and valuable. In this study the authors developed criteria for assessing the reliability of various published and unpublished SNP markers associated with several Pc genes in oats and validated it further by developing PACE assays. The study is useful contribution and resource for oat breeders for conducting reliable marker assisted selection and germplasm characterization for resistance to Pc. The manuscript is very well written and is of high standard. The objective of the study is clear, and methodology used is sound. Results are well presented, discussed and concluded.

I recommend the manuscript for publication after minor revisions (below):

1) The authors mentioned ‘this study considered a false positive rate of 25% or higher to

be excessively high'. What’s the basis of deciding on this benchmark? Clarification needed in support of this statement.

2) Table 1: Revise the values in rows under the column 'False positive rate (%)'. Remove ‘to’ and replace by hyphen (8-27, 23-63…).

3) Table 2 and 3 can be merged with extra column <10% and 10-25%.

Reviewer #2: Chowdhury reports on the validation of PACE assays interrogating SNPs linked to Pc genic information in the literature. The study is of value to the community however in my opinion 25 % false positive rate is far too high and the utility of many of these markers is overstated from a breeding perspective.

Why was the validation presented after the screening of the 92 susceptible oat lines?

Is it more logical to test the markers first (or present the validation first) in known carriers and non-carriers?

In the report, 20 of these 31 non-carrier lines in the validation panel were also part of the 92 lines - does this set function as negative control?

In the validation panel (S7), 12 of the 19 carrier lines were also used as positive controls, so again, makes more sense to report the 'validation' first, unless the final validation is performed in entirely new set of materials.

How reliable is the "non-carrier" status of 92 lines? Have these been genotyped/pedigreed? There are some lines in this panel that have unknown parentage. What are the chances of type I error to occur? Maybe elaborate this a little bit in the discussion.

Minor comments to address

Line 152 - Pc48 data not shown in S4 Table.

Line 167 - KASP genotyping- Seems to came out of the blue - was KASP genotyping also used?

Line 218 - Please clarify "in combination". What assay design was implemented? Was this described in Materials and Methods?

**Do you want your identity to be public for this peer review?** For information about this choice, including consent withdrawal, please see our Privacy Policy

Reviewer #1: **Yes: ** Davinder Singh

Reviewer #2: No

---

## [Author Response · Author response to Decision Letter 1]

12 Mar 2025

Academic editor

Response:

The manuscript has been checked whether the manuscript meets PLOS ONE's style requirements

“The research at USDA-ARS in Aberdeen, ID was supported with base fund Project No.: 2050-21000-034-000D”

Response:

The amended Role of Funder statement has been added to the cover letter.

3. When completing the data availability statement of the submission form, you indicated that you would make your data available on acceptance. We strongly recommend all authors decide on a data sharing plan before acceptance, as the process can be lengthy and hold up publication timelines. Please note that, though access restrictions are acceptable now, your entire data will need to be made freely accessible if your manuscript is accepted for publication. This policy applies to all data except where public deposition would breach compliance with the protocol approved by your research ethics board. If you are unable to adhere to our open data policy, please kindly revise your statement to explain your reasoning and we will seek the editor's input on an exemption. Please be assured that, once you have provided your new statement, the assessment of your exemption will not hold up the peer review process.

Response:

All relevant data are within the paper and its Supporting Information files.

Response:

We have included a separate supplementary table (S8 Table) containing all data that were not shown previously.

5. Please review your reference list to ensure that it is complete and correct. If you have cited papers that have been retracted, please include the rationale for doing so in the manuscript text or remove these references and replace them with relevant current references. Any changes to the reference list should be mentioned in the rebuttal letter that accompanies your revised manuscript. If you need to cite a retracted article, indicate the article’s retracted status in the References list and also include a citation and full reference for the retraction notice.

Response:

Citation number 22 (line 110) and citation number 28 (line 313) has been added to the manuscript to address reviewer’s comment.

22. Tinker NA, Chao S, Lazo GR, Oliver RE, Huang YF, Poland JA, et al. A SNP genotyping array for hexaploid oat. The Plant Genome. 2014;7(3):plantgenome2014.03.0010.

28. Bernardo AN, Bowden RL, Rouse MN, Newcomb MS, Marshall DS, Bai G. Validation of molecular markers for new stem rust resistance genes in US hard winter wheat. Crop Science. 2013;53(3):755-64.

Reviewer(s)

Reviewer #1:

The authors mentioned ‘this study considered a false positive rate of 25% or higher to be excessively high'. What’s the basis of deciding on this benchmark? Clarification needed in support of this statement.

Response:

Thank you for your comment. We have addressed this in the discussion section (lines 308 – 318). We agree that markers with low false positive rates are the best for breeding programs. Accordingly, this manuscript reports markers for three Pc genes (Pc38, Pc68 and Pc53 – Table 2) for breeding purposes as they have low false positive rates ranging from 0-10%. However, markers for the remaining genes (Table 2) with a false positive rate of 10-25% may find application in fine-mapping or specific populations. Oat breeders can choose markers and decide on the acceptable error rate based on the specific objectives of their breeding program. We have included previously published articles supporting our threshold level such as Zhao et al. (2020), Bernardo et al. (2013) (lines 308 – 318).

2) Table 1: Revise the values in rows under the column 'False positive rate (%)'. Remove ‘to’ and replace by hyphen (8-27, 23-63…).

Response:

Table has been corrected in the revised manuscript

3) Table 2 and 3 can be merged with extra column <10% and 10-25%.

Response:

Table 2 and 3 are merged in the revised manuscript

Reviewer #2:

1.Chowdhury reports on the validation of PACE assays interrogating SNPs linked to Pc genic information in the literature. The study is of value to the community however in my opinion 25 % false positive rate is far too high and the utility of many of these markers is overstated from a breeding perspective.

Response:

Thank you for your comment. We have addressed this in the discussion section (lines 308 – 318). We agree that markers with low false positive rates are the best for breeding programs. Accordingly, this manuscript reports markers for three Pc genes (Pc38, Pc68 and Pc53 – Table 2) for breeding purposes as they have low false positive rates ranging from 0-10%. However, markers for the remaining genes (Table 2) with a false positive rate of 10-25% may find application in fine-mapping or specific populations. Oat breeders can choose markers and decide on the acceptable error rate based on the specific objectives of their breeding program. We have included previously published articles supporting our threshold level such as Zhao et al. (2020), Bernardo et al. (2013) (lines 308 – 318).

2. Why was the validation presented after the screening of the 92 susceptible oat lines?

Is it more logical to test the markers first (or present the validation first) in known carriers and non-carriers? In the report, 20 of these 31 non-carrier lines in the validation panel were also part of the 92 lines - does this set function as negative control? In the validation panel (S7), 12 of the 19 carrier lines were also used as positive controls, so again, makes more sense to report the 'validation' first, unless the final validation is performed in entirely new set of materials.

Response:

Thank you for your thoughtful comment. We conducted validation after screening the 92 susceptible (non-carrier) oat lines because our primary goal was to identify potential candidate markers which has a potential to be used in marker-assisted selection. To ensure these markers were not linked due to random associations or population structure rather than true causal relationships, we estimated the false positive rate and used it to refine marker selection. The selected markers were then converted into a different marker system (PACE assays) and validated using a panel of carrier and non-carrier lines. This step allowed us to assess marker performance across a broader genetic background before testing their utility in known carriers and non-carriers.

We acknowledge that the validation panel included lines overlapping with the initial marker evaluation panel. Specifically, the 20 non-carrier lines in the validation panel that overlapped with the initial 92 lines were not explicitly designated as negative controls but provided a useful reference for cross-validation. Similarly, the 12 carrier lines present in both the initial screening and validation panel served as a consistency check rather than formal positive controls.

4. How reliable is the "non-carrier" status of 92 lines? Have these been genotyped/pedigreed? There are some lines in this panel that have unknown parentage. What are the chances of type I error to occur? Maybe elaborate this a little bit in the discussion.

Response:

The 92 non-carrier lines were genotyped using both the Illumina 6K and 3K (USDA_SoywheOatBar-3K) chips at the USDA–ARS Genotyping Laboratory in Fargo, ND, following the methodology described by Tinker et al. (2014) (lines 107-110). Additionally, their non-carrier status was verified through pedigree records in the Pedigrees of Oat Lines database (https://triticeaetoolbox.org/POOL/index_db.php). Furthermore, we have included a discussion section addressing the reliability of the non-carrier status of these 92 oat lines (lines 340-349).

5. Line 152 - Pc48 data not shown in S4 Table.

Response:

Data for Pc48 has been added in S4 Table.

6. Line 167 - KASP genotyping- Seems to came out of the blue - was KASP genotyping also used?

Response:

KASP genotyping was a mistake in typing, and it has been corrected as PACE genotyping

7. Line 218 - Please clarify "in combination". What assay design was implemented? Was this described in Materials and Methods?

Response:

The method used to determine the false positive rate (%) of the SNP combination has been added to the material and method section (lines 145-148).

---

## [Decision Letter · Decision Letter 1]

Development and accuracy assessment of molecular markers associated with crown rust resistance genes in oat

PONE-D-24-42419R1

Dear Dr. Chowdhury,

We’re pleased to inform you that your manuscript has been judged scientifically suitable for publication and will be formally accepted for publication once it meets all outstanding technical requirements.

Kind regards,

Dragan Perovic, Ph.D

Academic Editor

PLOS ONE

Additional Editor Comments (optional):

Dear Dr. Chowdhury,

It is my pleasure to accept your manuscript, "Development and Accuracy Assessment of Molecular Markers Associated with Crown Rust Resistance Genes in Oat," for publication, following the revisions made by you and your team.

With kind regards,

Dragan

Reviewers' comments:

Reviewer's Responses to Questions

**Comments to the Author**

Reviewer #1: All comments have been addressed

Reviewer #2: All comments have been addressed

2. Is the manuscript technically sound, and do the data support the conclusions?

Reviewer #1: Yes

Reviewer #2: Yes

3. Has the statistical analysis been performed appropriately and rigorously?

Reviewer #1: Yes

Reviewer #2: N/A

4. Have the authors made all data underlying the findings in their manuscript fully available?

Reviewer #1: (No Response)

Reviewer #2: Yes

5. Is the manuscript presented in an intelligible fashion and written in standard English?

Reviewer #1: Yes

Reviewer #2: Yes

Reviewer #1: (No Response)

Reviewer #2: Thank you for considering my comments in the review of the manuscript. The manuscript is much improved and suitable for publication

**Do you want your identity to be public for this peer review?** For information about this choice, including consent withdrawal, please see our Privacy Policy

Reviewer #1: **Yes: ** Davinder Singh

Reviewer #2: No

---

## [Editor Report · Acceptance letter]

PONE-D-24-42419R1

PLOS ONE

Dear Dr. Chowdhury,

I'm pleased to inform you that your manuscript has been deemed suitable for publication in PLOS ONE. Congratulations! Your manuscript is now being handed over to our production team.

Kind regards,

on behalf of

Dr. Dragan Perovic

Academic Editor

PLOS ONE